



# Characteristics of bacterial and fungal communities and their associations with sugar compounds in atmospheric aerosols at a rural site in North China

Mutong Niu[1], Shu Huang[1], Wei Hu[1, 2, *], Yajie Wang[1], Wanyun Xu[3], Wan Wei[1], Qiang Zhang[1], Zihan Wang[1], Donghuan Zhang[1], Rui Jin[1], Libin Wu[1], Junjun Deng[1], Fangxia Shen[4], Pingqing Fu[1, *]

[1]Institute of Surface-Earth System Science, School of Earth System Science, Tianjin University, Tianjin 300072, China
[2]Tianjin Bohai Rim Coastal Earth Critical Zone National Observation and Research Station, Tianjin University, Tianjin 300072, China
[3]State Key Laboratory of Severe Weather & Key Laboratory for Atmospheric Chemistry, Institute of Atmospheric Composition, Chinese Academy of Meteorological Sciences, Beijing, China
[4]School of Space and Environment, Beihang University, Beijing 102206, China

*Correspondence to*: W. Hu (huwei@tju.edu.cn), P. Fu (fupingqing@tju.edu.cn)

**Abstract.** Bioaerosols play significant roles in causing health and climate effects. Sugar compounds in air have been widely used to trace the source of bioaerosols. However, knowledge about the association of sugar molecules and the microbial community at taxonomic levels in atmospheric aerosols remains limited. Here, microbial community compositions and sugar molecules in total suspended particles collected from a typical rural site, Gucheng, in the North China Plain were investigated by gas chromatography-mass spectrometry and high-throughput gene sequencing, respectively. Results show that fungal community structure exhibited distinct diurnal variation with largely enhanced contribution of Basidiomycota at night, while bacterial community structure showed no obvious difference between daytime and night. SourceTracker analysis revealed that bacteria and fungi were mainly from plant leaves and unresolved sources (presumably human-related emission and/or long-distance transport), respectively. All the detected anhyrosugars and sugar alcohols, and trehalose showed diurnal variations with lower concentrations in the daytime and higher concentrations at night, which may be affected by enhanced fungal emissions at night, while primary sugars (except trehalose) showed an opposite trend. The Mantel test resulted that more sugar compounds exhibited significant associations with fungal community structure than bacterial community structure. Co-occurrence analysis revealed the strong associations between sugar compounds and a few saprophytic fungal genera with low relative abundances, e.g., *Hannaella*, *Lectera*, *Peniophora*, *Hydnophlebia*, *Sporobolomyces* and *Cyphellophora*. This study suggested that the entire fungal community likely greatly contributed to sugar compounds in rural aerosols, rather than specific fungal taxa, while the contribution of bacteria was limited.



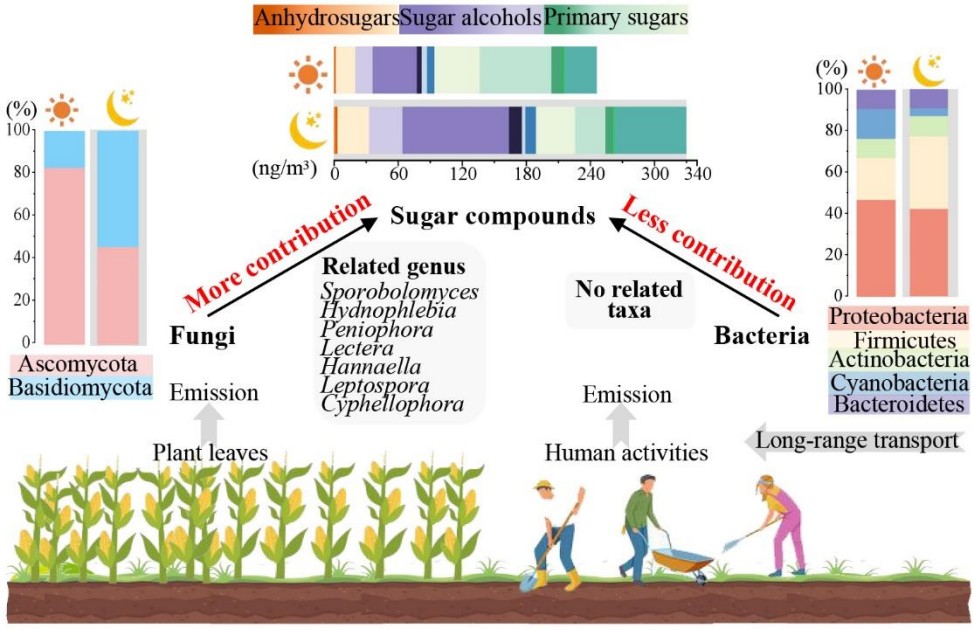



## 1 Introduction

Bioaerosols, mainly refer to particulate matter with biological origin, including viruses, bacteria, fungi, pollen and their biological debris (Yao, 2018;Fröhlich-Nowoisky et al., 2016;Santl-Temkiv et al., 2020;Shen and Yao, 2022). Bioaerosols have attracted progressively more attention during the past decades due to their adverse health effects. Health problems can arise in

living organisms such as humans, animals and plants through respiratory or surface-contact exposure to airborne pathogenic or allergenic microbes (Taipale et al., 2021;Arzt et al., 2011;Hicks et al., 2012), which have caused serious economic losses and even human deaths. Bioaerosols also play potentially important roles in cloud and precipitation formation by acting as cloud condensation nuclei and ice nucleating particles (Hoose et al., 2010;Ariya et al., 2009;Hu et al., 2020;Huang et al., 2021). Certain airborne microorganisms can participate in atmospheric chemical processes by transforming inorganic or organic

matter (Liu et al., 2020;Ariya et al., 2002).

In order to accurately assess the health-related and climatic effects of bioaerosols, and more importantly, to develop effective bioaerosol pollution control approaches, the investigations on the abundance, compositions and emission fluxes of bioaerosols need to be taken seriously. Previous studies have found that in terms of number and mass concentrations, bioaerosols could account for ~30% of total aerosols (size larger than ~1 μm) in urban or rural sites (Després et al., 2012;Huffman et al.,

2013;Schumacher et al., 2013); while in rainforest, bioaerosols account for up to 80% (Huffman et al., 2012;Pöschl et al., 2010). Current estimates of global bioaerosol mass emission fluxes vary widely (about 10–1000 Tg a$^{-1}$) (Després et al., 2012). In recent decades, molecular biology technology (e.g., fluorescence in situ hybridization techniques, denaturing gradient gel electrophoresis, terminal restriction fragment length polymorphism and sequencing technology) and fluorescence techniques (e.g., ultraviolet aerodynamic particle sizer and wideband integrated bioaerosol sensor) have been applied to study the

compositions and abundances of bioaerosols (Prass et al., 2021;Drautz-Moses et al., 2022;Zhang et al., 2017). Whereas, the compositions, sources and emission fluxes of bioaerosols remain less documented and quantified (Yue et al., 2016;Fröhlich-Nowoisky et al., 2016).

Sugar compounds are the main energy storage and cellular structural materials in organisms (Wang et al., 2021), and their presence in the atmosphere is often considered to be closely related to biological sources (Bauer et al., 2008a;Schliemann et

al., 2008). Hence sugar compounds are often used as biomarkers to track natural sources of bioaerosols and estimate the contribution of bioaerosols to organic aerosols (Fu et al., 2010;Fu et al., 2013;Fu et al., 2008). For instance, arabitol and mannitol have been commonly used to assess fungal spore emissions (Bauer et al., 2008a;Marynowski et al., 2019), and trehalose was considered a fungal-specific sugar since it is mainly present in mycorrhizal plants (Kaur and Suseela, 2020). Sucrose and glucose, as end products of photosynthesis, were often used to trace the emissions from plant debris and some

photosynthetic prokaryotes (Goddijn and van Dun, 1999). Fructose can be used to study the spread of plant debris or pollen in the atmosphere (Fu et al., 2013).

In fact, however, most sugar compounds are widely present in living organisms. For example, a variety of bacteria have been found to synthesize mannitol (Monedero et al., 2010). Trehalose participates in osmotic protection and maintains desiccation



tolerance in bacteria, yeast, insects and some plants (Zancan and Sola-Penna, 2005). Glucose, fructose, galactose and mannose
were widely found in bacteria (Walmsley et al., 1998; Deutscher et al., 2006). That is, although the sugar compounds used as
biomarkers tend to be predominant in organisms they characterize, there are many possible biological sources of these sugar
compounds.

A large number of studies using sugar compounds as biomarkers focused on bacteria and fungi at the kingdom level, which
may result in inaccurate source identification and emission flux estimation of bioaerosols. To the best of our knowledge, only
Samaké et al. (2020) explored the relationship between sugar compounds and microorganisms in $PM_{10}$ collected in an
agricultural area in France, and there is still a large lack of understanding of the relationship between microbial communities
and sugar compounds at the molecular level. Such information is important for clarifying the sources of bioaerosols and
improving the accuracy of bioaerosol emission flux estimates.

In this study, to refine the marker categories of sugar compounds in atmospheric aerosols, airborne total suspended particles
(TSP) were collected from a typical rural site, Gucheng, in the North China Plain. Atmospheric microbial communities
(bacteria and fungi) and common sugar compounds in the TSP samples were analyzed using high-throughput gene sequencing
and gas chromatography-mass spectrometry (GC-MS) techniques, respectively. Statistical analyses were exploited to search
for the association between microbial communities and sugar compounds in atmospheric aerosols.

## 2 Materials and methods

### 2.1 Observation and sample collection

The observation was carried on at the rooftop of the Gucheng Ecological and Agrometeorological Experiment Station, the
Chinese Academy of Meteorological Sciences (39.15°N, 115.73°E; about 5 m above the ground level) in the summer of 2020
(7−13 August). The observation site is located in Dingxing County, Hebei Province, which is about 100 km southwest away
from Beijing and 128 km west away from Tianjin (**Fig. S1**). The station is situated in the high-yield agricultural area in the
north of the North China Plain, where the main crop in summer is maize. TSP samples loaded on quartz fiber filters ($25 \times 20$
cm) were collected using a high-volume air sampler (Tisch Environmental, Inc., USA) daily based on daytime (D, 8:00–19:30)
and nighttime (N, 20:00–7:30) at a flow rate of 1.05 $m^3\ min^{-1}$. All filters before sampling were combusted at 450℃ for 6 h to
remove organic matter and microorganisms. The filters after sampling were stored in a refrigerator at −20℃ for later analysis.
Meteorological parameters including air temperature, relative humidity, wind speed and direction, and precipitation are real-
time monitoring data obtained from a weather station in Xushui, Hebei (about 18 km away from the sampling site). Air quality
parameters including AQI, $PM_{2.5}$, $PM_{10}$, $NO_2$, $O_3$, $SO_2$ and CO at the Baoding environmental monitoring station (about 36 km
away from the sampling site) were obtained from China National Environmental Monitoring Center (http://www.cnemc.cn/).



## 2.2 Taxonomic identification of airborne microbial community

High-throughput gene sequencing technique was used to analyze airborne bacterial and fungal community structures. A piece
of 18 cm$^2$ filter for each sample was cut and placed in lysis tube, added to Solution 1 provided by PowerSoil DNA Isolation
Kit (MoBio Laboratories, USA) and then placed at 65℃ for 20 min to reduce the adsorption of quartz to DNA. Subsequently,
the solution was processed according to the manufacturer's instructions. Polymerase chain reaction (PCR) amplification was
conducted in a 25 μL reaction solution, including 12.5 μL KAPA 2G Robust Hot Start Ready Mix, 1 μL of 5 μM of each
primer, 1.5 μL genome DNA and 9 μL double distilled H$_2$O. The PCR primer pairs are 338F (5'-
ACTCCTACGGGAGGCAGCA-3') and 806R (5'-GGACTACHVGGGTWTCTCATAT-3') in bacterial V3-V4 region, and
ITS1F (5'-CTTGGTCATTTAGAGGAAGTAA-3') and ITS2 (5'-TGCGTTCTTCATCGATGC-3') in fungal ITS region.
The PCR amplification was performed as follows: denaturation at 94℃ for 5 min, 30 cycles for bacteria and 35 cycles for
fungi of denaturation, annealing and extension (94℃ for 30 s, 50℃ for bacteria and 55℃ for fungi for 30 s, and 72℃ for 60
s), and final extension at 72℃ for 7 min. The final PCR products were purified by AMPure XP magnetic beads. Then the
purified DNA was used for generating a sequencing library. The library was purified by AMPure XP magnetic beads and
qualified with NanoDrop 2000 Spectrophotometers (Thermo Fischer Scientific, Inc., USA). Finally, the library was sequenced
using the Illumina Miseq 2500 platform (2×250 paired ends) at Allwegene Company, Beijing, China.
The pair-end reads were quality-controlled using Trimmomatic v0.33, and then merged using FLASH v1.2.11 software to
produce the raw tags. Subsequently, the raw tags were identified based on the barcode and primer using Mothur v1.35.1 and
were filtered by removing the barcode and primer. Then the clean sequences with ≥ 97% similarity were assigned to one
operational taxonomic unit (OTU). Taxonomic annotation of each OTU for bacteria and fungi was performed using the Silva
16S and UNITE databases, respectively.

## 2.3 Determination of sugar components

A piece of each sampled filter was extracted by sonication for 10 min using a 10 mL dichloromethane and methanol mixture
(2:1, v/v). The extracts were filtered through Pasteur pipette filled with quartz wool and concentrated to approximately 100 μL
using a rotary evaporator, and then dried under nitrogen. The silylation derivatization was performed by reaction with 60 μL
*N, O*-bis-(trimethylsilyl) trifluoroacetamide (BSTFA) including 1% trimethylsilyl chloride and pyridine (5:1, v/v) at 70 °C for
3 h. After the derivatization, hexane solution containing C$_{13}$ n-alkane internal standard (1.43 ng/μL) was added to the
derivatives. The samples were stored at −20℃ before GC-MS analysis.
Sugar compounds were quantified by a GC-MS system (7890A-5975C, Agilent Technology, USA). The GC was equipped
with a split/splitless injection and a fused silica capillary column (DB-5MS, 30 m × 0.25 mm × 0.25 μm, Hewlett-Packard).
The GC oven temperature procedure was as follows: initiation at 50℃ for 2 min, heating to 120℃ at a rate of 15℃/min,
heating to 300℃ at a rate of 5℃/min, stabilization for 16 min, and finally decreasing to 50℃. The carrier gas was ultrapure
helium (1.3 mL/min). The GC injector temperature was 280℃, the ion source temperature was 230℃, and the quadrupole

temperature was 150°C. The MS was operated on Electron Ionization (EI) mode at 70 eV and scanned from 50 to 650 Da.
Mass spectral data were processed using the ChemStation software, and sugar compounds were identified and quantified using authentic standards.

## 2.4 Chemical components analyses

Organic (OC) and elemental carbon (EC) were quantified by an OC/EC analyzer (Sunset Laboratory Inc., USA) according to
the thermal optical transmission (TOT) method of the NOISH protocol.

A portion of the filters was cut and extracted with 20 mL ultrapure water by ultrasonication for 20 min. Then the extracts were filtered using 0.22 m PTFE filters (MillexGV, Millipore, USA). The filtrate was used for the determination of water-soluble inorganic components including $Na^+$, $K^+$, $Mg^{2+}$, $Ca^{2+}$, $NH_4^+$, $SO_4^{2-}$, $NO_3^-$, and $Cl^-$ with an ions chromatography system (ICS-5000+, Thermo Fisher, USA).

## 2.5 Statistical analyses

The source samples used as source tracking reference databases in this study were soils, plant leaves and seawater. Maize leaves and soil samples were collected around the observation site and the samples were subjected to high-throughput sequencing as described in Section 2.2 to obtain the gene sequences, and the marine sequences were obtained from the NCBI SRA database (https://www.ncbi.nlm.nih.gov/sra/, **Table S1**). These reference sequences were analyzed together with aerosol
sample sequences by QIIME 2 to obtain OTU tables. The source tracker package "SourceTracker" for the R software was used for source tracking analysis of the airborne microbial community.

The paired-sample t-test was performed using SPSS v.22.0 to analyze the diurnal differences in the microbial community. Spearman correlation analysis was used to evaluate the relationship between microbial taxa and environmental or pollution factors. A $p$-value $< 0.05$ indicates a significant correlation. Non-metric multidimensional scaling (NMDS), principal
coordinate analysis (PCoA), redundancy analysis (RDA) and the Mantel test were conducted using the R scripts. The data needed for the co-occurrence analysis was processed through the R script and the results are visualized through Gephi v0.9.3.

## 3 Results and Discussion

### 3.1 Diurnal variation in microbial community

#### 3.1.1 Microbial diversity and richness

A total of 38565 16S rRNA gene sequences and 56920 ITS sequences were obtained by high-throughput sequencing technology and were assigned to 404 bacterial OTUs and 391 fungal OTUs. The sequencing coverage of all the samples was > 99%, indicating that the sequencing data could represent the microbial communities in the atmosphere.



Alpha diversity indices were used to evaluate the diurnal differences in microbial community structure (**Fig. 1** and **Table S2**). For bacterial communities, OTU and the Chao1 index used to evaluate richness showed no significant diurnal patterns. Except for the samples collected on August 13, the bacterial diversity (Shannon and Simpson's indices) in the daytime samples was slightly lower or comparable to that in the nighttime samples ($p > 0.05$, **Fig. 1a**). In contrast, the diversity and richness of fungal community were significantly higher at night than during the daytime ($p < 0.05$, **Fig. 1b**). Abdel Hameed et al. (2009) found that bacteria and fungi had similar diurnal variation patterns over Cairo City, Egypt. However, the bacterial diversity in aerosols over Guilin, China was lightly abundant during the day than at night (Long et al., 2022), which was similar to the finding over tropical forest aerosols in August (Gusareva et al., 2019). These different results suggest that the diversity and biogeography of airborne microorganisms might be influenced by multiple factors, e.g., geographical locations, airflow patterns, and global atmospheric circulation (Fröhlich-Nowoisky et al., 2012;Martiny et al., 2006;Womack et al., 2010).

### 3.1.2 Bacterial community structure

**Fig. 2** shows the bacterial and fungal community compositions in TSP samples collected at rural Gucheng, North China. At the phylum level, 38 bacterial phyla were detected. Proteobacteria (42.2%), Firmicutes (26.1%), Actinobacteria (9.0%), Cyanobacteria (8.7%) and Bacteroidetes (8.4%) were the most abundant bacterial phyla among all the TSP samples (**Fig. 2a**), which were also observed as the dominant phyla in the air of other regions (Acuna et al., 2022;Chen et al., 2022;Xu et al., 2017). The relative abundance of Cyanobacteria slightly increased in the daytime ($p > 0.05$), which was more abundant in the daytime with warmer temperatures (Singh et al., 2018). Firmicutes had a higher relative abundance in the nighttime ($p < 0.05$, **Fig. 3a**). Gusareva et al. (2019) also found that high temperature had an important impact on the bacterial community structure, and the abundance of Firmicutes was higher at noon when the temperature was the highest. Drautz-Moses et al. (2022) reported significant diurnal differences in the relative abundance of the phylum Firmicutes in aerosols in the lower troposphere.

At the order level, Bacillales (15.1%), Clostridiales (9.6%), Burkholderiales (8.8%), Sphingomonadales (5.8%) and Myxococcales (5.4%) were predominant (**Fig. S2a**) and no significant diurnal differences were found at the order level. At the genus level, *Bacillus* (6.9%), *Curvibacter* (6.3%), *Chroococcidiopsis* (4.2%) and *Tumebacillus* (4.1%) were the most abundant bacterial genera (**Fig. 2b**). *Bacillus* and *Chroococcidiopsis* were the dominant bacterial genera in the air worldwide (Zhao et al., 2022). *Curvibacter* occurring mainly in the aqueous environment (Xu et al., 2021) and spore-forming bacterium *Tumebacillus* occurring in the permafrost (Steven et al., 2008) and soils (Wemheuer et al., 2017) had smaller relative abundances observed in previous studies (Yan et al., 2018). Geographical locations and land use types may shape atmospheric bacterial community structure (Bowers et al., 2011). Shen et al. (2019) found that the relative abundances of *Bacillus* and *Curvibacter* in rural aerosols were about 90 times higher than those in urban aerosols. Based on the results of NMDS analysis, the diurnal difference in bacterial community structure was not apparent, while fungal community structure exhibited distinct diurnal differences (**Fig. S3**).





### 3.1.3 Fungal community structure


Fungal communities were almost always composed of the phyla Ascomycota (63.1%) and Basidiomycota (36.7%) (**Fig. 2c**). Their relative abundances were similar to the results of rural aerosols collected from Backgarden, South China (Fröhlich-Nowoisky et al., 2012). Due to the more abundant Basidiomycota found in natural soils (Hui et al., 2017), Basidiomycota accounted for a relatively higher proportion of rural aerosols compared with urban aerosols (Liu et al., 2019). Significant

diurnal differences were found in these two phyla (daytime: Ascomycota (65.3%) vs Basidiomycota (13.6%), nighttime: Ascomycota (28.6%) vs Basidiomycota (34.0%), $p < 0.01$) (**Fig. 3b**). The release of fungal spores is strongly linked to environmental humidity (Niu et al., 2021). It has been found that Ascomycota prefers to release spores in relatively dry environments, while Basidiomycota prefers wet environments (Almaguer et al., 2014;Elbert et al., 2007). Drautz-Moses et al. (2022) also observed significant diurnal differences in certain fungal phyla, e.g., Basidiomycota had higher proportions at

night. At the order level, Pleosporales (67.4%) dominated in the daytime, while Pleosporales (33.5%), Agaricales (34.6%) and Polyporales (13.3%) were the dominant orders in the nighttime (**Fig. S2b**). Polyporales and Agaricales, belonging to Basidiomycota, were observed to enhance in autumn when rainfall is higher (Abrego et al., 2018). These saprotrophic Basidiomycetes could release more spores at night due to higher air moisture (Kramer, 1982).

Significant diurnal variation was also observed for the fungal genera (**Fig. 2d**). *Coprinopsis*, *Coprinellus* and *Trametes* were

dominant in all the samples, and their abundances were significantly higher at night than in the daytime (*Coprinopsis*: 29.9% > 6.9%, *Coprinellus*: 11.2% > 8.9% and *Trametes*: 7.7% > 5.8%). The abundances of *Aspergillus* (10.1%), *Periconia* (8.4%) and *Cladosporium* (5.5%) increased significantly in the daytime (**Fig. 3b**). Similar diurnal patterns were found in the studies of airborne fungal spores in Valladolid (Spain) (Reyes et al., 2016) and Oklahoma (USA) (Gillum and Levetin, 2008). Air relative humidity (RH) facilitates the sporulation process, accompanied by a reduction of atmospheric turbulence (Oliveira et

al., 2009;Drautz-Moses et al., 2022), so fungal community structure showed more significant diurnal variations than bacteria.

### 3.2 Possible factors affecting microbial community

Previous studies have shown that airborne microbial community structure may be influenced by synergistic effects of geographic, meteorological and pollution factors (Zhai et al., 2018;Zhao et al., 2022). To analyze the factors affecting the microbial community at the typical rural site, Gucheng, the influences of geographic, meteorological and pollution factors on

the variations in microbial community structure were investigated in this study. As mentioned above, Gucheng rural site in the North China Plain is geographically close to the megacities of Beijing and Tianjin, but the microbial community composition in Gucheng rural aerosols was significantly different from those in urban Beijing and Tianjin aerosols (**Fig. S4**). The analysis based on the SourceTracker method revealed that the airborne fungi at Gucheng site were mainly derived from terrestrial plants (44.1 ± 29.1%), while the sources of bacteria were mostly unresolved (98.9 ± 1.0%) (**Fig. 4a**). The results in this study are

similar to the previous view that plant leaves have a greater contribution to local airborne fungi (Qi et al., 2020).



It should be noted that only the contribution of natural sources including terrestrial plants, soils and ocean to the airborne microbial community was considered in this study, and unresolved sources may be human-related, such as human body surface and human gut (feces), which have been shown to be major sources of airborne bacteria (Jiang et al., 2022;Zhao et al., 2022). It is also noticeable that in this study only local soil and vegetation samples were selected as the source samples, and the impact

of long-distance transport was not considered. Therefore, unresolved sources may also come from long-range transported air masses, although local sources are often considered to play leading roles in shaping microbial communities (Zhai et al., 2018;Bowers et al., 2011).

To explore the relationship between microbial communities and environmental factors, RDA analysis was performed. As shown in **Fig. 4b**, $Ca^{2+}$ and wind speed (WS) were the main environmental variables affecting the bacterial community

structure. $Ca^{2+}$ is an important tracer of soil dust particles (Mace et al., 2003). The results implied that the emission from soils played an important role in shaping the bacterial community. High WS is conducive to the resuspension of soil microorganisms, bringing soil microorganisms into the atmosphere (Savage et al., 2012;Jones and Harrison, 2004). In addition, the long-distance transport of air masses is often accompanied by high WS, which may carry microorganisms and affect the bacterial community structure (Zhai et al., 2018).

Air temperature, RH, and ozone ($O_3$) were the vital factors affecting the fungal community structure (**Fig. 4b**). Temperature and RH could affect the growth and release of microorganisms from the sources, especially fungi. A large number of studies have proved that temperature and RH are able to interfere with airborne fungal communities (Wang et al., 2020;Núñez et al., 2021). Different temperatures can activate the formation and release of different types of fungal spores (Pickersgill et al., 2017). High RH could promote the release of many types of basidiospores and ascospores (Elbert et al., 2007), while when the RH

decreases to 40–60%, the release of conidia becomes easier (Jones and Harrison, 2004). The oxidation by ozone can effectively inhibit the growth of microorganisms (Zhai et al., 2018). In contrast, Grinn-Gofroń et al. (2011) found that there were significant positive correlations between $O_3$ and fungal spores, while Sousa et al. (2008) suggested that $O_3$ had little impact on fungi. It is presumed that the influence of ozone on fungi depends on the type of fungi or the joint effect of related environmental factors.

**3.3 Diurnal variation in sugar compounds**

A total of 13 sugar compounds including three anhyrosugars, five primary sugars and five sugar alcohols were analyzed in this study. The total concentration of sugar compounds in all the TSP samples averaged $289 \pm 85.3$ ng m$^{-3}$ (**Fig. 5a**), which was comparable to the previously reported concentrations in rural and mountain aerosols (Yan et al., 2019;Li et al., 2012). The daytime concentration of sugar compounds was $247 \pm 67.0$ ng m$^{-3}$, which was lower than that ($331 \pm 84.4$ ng m$^{-3}$) in the

nighttime (**Table S3**).



### 3.3.1 Anhyrosugars

Anhyrosugars (levoglucosan and its isomers, galactosan and mannosan) are the main constituents of cellulose (Simoneit, 2002) and are specific markers for biomass burning (Simoneit et al., 1999). The concentrations of levoglucosan were in the range of 10.3–30.3 ng m$^{-3}$ (average: 18.4 ± 7.43 ng m$^{-3}$) in the daytime and 2.97–55.8 ng m$^{-3}$ (average: 30.0 ± 20.1 ng m$^{-3}$) in the nighttime, showing a diurnal variation. The abundance of levoglucosan was about 10–30 times higher than those of galactosan (daytime: 0.7 ± 0.1 ng m$^{-3}$, nighttime: 1.4 ± 1.0 ng m$^{-3}$) and mannosan (daytime: 1.3 ± 0.5 ng m$^{-3}$, nighttime: 2.1 ± 1.6 ng m$^{-3}$) (**Table S3**). Anhyrosugars concentrations exhibited diurnal variations (**Fig. 5a**), which may be influenced by cooking activities (Verma et al., 2021). Yi et al. (2021) also reported that the anhyrosugars concentrations in PM$_{2.5}$ were lower in the daytime (Tai´an: 52.3 ± 36.3 ng m$^{-3}$, Mt. Tai: 25.4 ± 16.2 ng m$^{-3}$) than at night (Tai´an: 55.8 ± 44.3 ng m$^{-3}$, Mt. Tai: 30.0 ± 19.5 ng m$^{-3}$) in Shandong Province, China. It is noteworthy that the anhyrosugars concentrations in this study decreased sharply compared to those over the same sampling site in the summer of 2013 (7.3–1599.6 ng m$^{-3}$) (Li et al., 2019). This decreases likely implies that biomass burning events were effectively controlled between 2013 and 2020 due to more strict air pollution control measures, e.g., banning of burning waste straw in agricultural areas.

### 3.3.2 Primary sugars

Primary sugars including fructose, glucose, sucrose, trehalose and xylose were detected in all the TSP samples. The primary sugars in the daytime and nighttime samples were 152 ± 49.0 ng m$^{-3}$ and 141 ± 47.2 ng m$^{-3}$, respectively. These primary sugars dominated the total sugar compounds in the daytime samples (61.3 ± 0.1%), and relatively lower proportions were found at night (42.1 ± 0.1%, **Table S3**), which was consistent with the results in previous studies (Zhao et al., 2020;Ren et al., 2018). Sucrose was the primary saccharide with the highest concentration in the daytime (67.6 ± 35.8 ng m$^{-3}$), followed by glucose (42.4 ± 14.7 ng m$^{-3}$). Sucrose has the potential origins of pollen (Fu et al., 2012) and soil resuspension (Rogge et al., 2007), and can be produced by photosynthesis in plant leaves (Medeiros et al., 2006b). However, no significant correlation was found between sucrose and soil resuspension-related indicators (Ca$^{2+}$ and PM$_{10}$). Sucrose had a significant positive correlation with xylose (**Table S4**), which mainly originates from wood and rice straw (Chen et al., 2013;Sullivan et al., 2011), indicating that sucrose and xylose may have a similar source. It is speculated that the elevated concentrations of sucrose in the daytime may be caused by the more active metabolic activity of plants.

Trehalose was the dominant primary saccharide at night with an average of 67.6 ± 23.4 ng m$^{-3}$, followed by glucose (36.6 ± 12.8 ng m$^{-3}$) (**Fig. 5a**). Trehalose exhibited a diurnal variation with higher concentrations at night and lower concentrations in the daytime, which was in contrast to the variations of the other four detected primary sugars (**Table S3**). Trehalose, as a reserve carbohydrate and pressure protector (Medeiros et al., 2006a), is widely found in various organisms, such as bacteria, fungi, insects, invertebrates, algae and higher plants (Medeiros et al., 2006b). As the most abundant sugar compound in soils, trehalose was used as the tracer for soil dust resuspension (Jia and Fraser, 2011;Rogge et al., 2007). Whereas, in this study, there was no significant correlation between trehalose and soil resuspension-related indicators (Ca$^{2+}$ and PM$_{10}$). Trehalose was



significantly negatively and positively correlated with air temperature and relative humidity, respectively (**Table S4**). Trehalose exhibited an apparent diurnal variation (**Fig. 5a**), which was consistent with the diurnal variation trend of fungal
OTUs (**Fig. 1b**). These results indicate that trehalose was possibly affected by fungal emissions.

### 3.3.3 Sugar alcohols

Arabitol, mannitol, erythritol, glycerol and inositol were the main sugar alcohols detected in the TSP samples. The total concentration of sugar alcohols was averaged $115.4 \pm 50.3$ ng m$^{-3}$ and their relative abundance was lower in the daytime ($30.6 \pm 0.07\%$) than that in the nighttime ($48.4 \pm 0.08\%$) (**Fig. 5a**). Mannitol was the predominant sugar alcohols in all the samples
($71.1 \pm 34.8$ ng m$^{-3}$), followed by arabitol ($23.6 \pm 12.5$ ng m$^{-3}$). Mannitol and arabitol were the most predominant organic compounds used to indicate fungal spores and showed significant correlations with fungal community diversity in terms of OTU abundance (**Table S4**).

The concentrations of mannitol and arabitol are commonly used to evaluate the contribution of fungal spores to ambient OC (Bauer et al., 2008a). The contributions of other primary emission sources such as biomass burning and plant debris to OC
could be estimated by levoglucosan and glucose, respectively (Fu et al., 2014;Puxbaum et al., 2007;Puxbaum and Tenze-Kunit, 2003;Andreae and Merlet, 2001). The results estimated based on these tracers show that fungal spore-derived OC dominated ($204.3-977.8$ ngC m$^{-3}$), accounting for 3.6–19.7% of OC. Plant debris and biomass burning contributed 0.4–1.6% and 1.1–6.9% to OC, respectively (**Fig. 5b**). Biological sources contributed more to primary OC in rural aerosols, while biomass-burning OC in urban aerosols were dominant in total OC (Wu et al., 2020;Yan et al., 2019). The nighttime fungal spore-derived
OC ($525.5-977.8$ ngC m$^{-3}$, 769.1 ngC m$^{-3}$ on average) was significantly higher than that in the daytime ($204.3-427.9$ ngC m$^{-3}$, 217.8 ngC m$^{-3}$ in average) ($p < 0.01$), which was consistent with the observation in other studies (Kang et al., 2018;Zhu et al., 2016). Many fungal spores are wet spores, preferring to release spores at night when the RH is higher.

### 3.4 Relationship between microbial community and sugar compounds

### 3.4.1 Correlation between microbial community structure and sugar compounds

To analyze the relationship between different microbial communities and sugar compounds, this study adopted the Mantel test to determine the possible correlation between microbial community structure and sugar compounds (**Fig. 6a**). The bacterial community structure was closely related to sucrose, while fungal communities were associated with more detected sugar molecules, including all the detectable anhydrosugars, two sugar alcohols mannitol and erythritol and one primary sugar sucrose. These results indicate that the fungal community may have a larger contribution to more sugar molecules than the
bacterial community in the rural atmosphere. Compared to bacteria in the air, fungal emissions and concentrations are lower in number, but much (an order of magnitude) higher in mass (Fröhlich-Nowoisky et al., 2016;Hoose et al., 2010). Anhyrosugars are important tracers of biomass burning (Simoneit et al., 1999) and sucrose is considered mainly from wood and rice straw. Fungi are more common on wood or plant surfaces than bacteria (Lladó et al., 2017;Andrews and Harris, 2000).



In this study, three genera *Coprinopsis*, *Coprinellus* and *Trametes* dominant in the fungal community, are typical wood
saprophytic fungi (Kamada et al., 2010;Nagy et al., 2012). Minor fungal genera, such as *Periconia* and *Cladosporium*, belong
to endophytes (Gunasekaran et al., 2021). As mentioned above, SourceTracker analysis revealed that fungi were mainly
derived from plants (**Fig. 4a**). Meanwhile, fungi can use levoglucosan or sucrose as the carbon source to promote growth
(Prosen et al., 1993;Van der Nest et al., 2015). It was also found that bacterial community structure may be closely related to
sucrose-traced plant origins, although the SourceTracker analysis results did not show the plant origin of bacteria. This
inconsistency may be caused by single plant (maize) source samples and/or the negligence of long-distance transport in this
study.

The correlations between the fungal community and two sugar alcohols (mannitol and erythritol) were the most significant
(correlation coefficients: 0.49 and 0.55) (**Fig. 6a**). Mannitol exists in fungal spores, fruiting bodies and mycelia of fungi
(Solomon et al., 2007), and is considered to be the most abundant dissoluble starch inside mycelia and fruiting bodies (Dulermo
et al., 2009). In fungi, mannitol can be used to store energy and extinguish reactive oxygen species (Upadhyay et al., 2015).
The natural sources of erythritol are mainly yeasts and yeast-like fungi, such as *Moniliella pollinis*, *Trichosporonoides
megachiliensis*, *Aureobasidium* sp., *Trigonopsis variabilis*, *Trichosporon* sp., *Torula* sp., and *Candida magnoliae* (Regnat et
al., 2018). They mainly produce erythritol when encountering salt or osmotic pressure (Yang et al., 2015). Mannitol has been
widely used to trace the origin of fungal spores in aerosols (Bauer et al., 2008b;Bauer et al., 2008a), and erythritol was
considered to have similar sources to mannitol (Chen et al., 2013).

### 3.4.2 Interaction between sugar molecules and microorganisms at taxonomic levels

To elucidate the interaction between sugar molecules and microorganisms at taxonomic levels, a co-occurrence network was
constructed based on Spearman's correlation between different microbial taxa and sugar molecules. All microbial taxa related
to the detected sugar molecules belong to fungi. The resulting fungal network consisted of 28 nodes and 27 edges, and was
divided into seven modules by modularizing the network (**Fig. 6b**), accounting for 28.6%, 17.9%, 14.3%, 14.3%, 10.7%, 7.1%
and 7.1% of all nodes, respectively.

Strong relationships can be observed between each module and different types of sugar molecules. Sugar alcohols used to trace
fungal spores were included in Module I and showed strong interactions with the order Cantharellales, the families
Cyphellophoraceae, Peniophoraceae and Ceratobasidiaceae, and the genera *Peniophora*, *Sporobolomyces* and *Cyphellophora*.
The members of Modules I, II and IV were mixed. In Module II, the class Microbotryomycetes, the family Sporidiobolaceae,
and the genera *Sporobolomyces* and *Hydnophlebia* were aggregated in erythritol. Module IV included trehalose which was
associated with the genera *Hannaella*, *Lectera* and *Sporobolomyces*. Model III only included the detectable anhyrosugars and
xylose. Modules V to VII was composed of only one primary sugar and a few fungal taxa. A previous study has shown that
there is a significant positive correlation between *Sporobolomyces* and sugar compounds (Samaké et al., 2020). Although the
fungal genera contained in the network are not the dominant fungi (0.007–0.22%), these genera are wood or litter saprophytic
fungi that may originate from vegetation, demonstrating the important influence of vegetation on sugar compounds in aerosols.





As mentioned above, a survey conducted by Samaké et al. (2020) at a rural site in France is the only study investigating the association between microorganisms and sugar compounds, which found that the genera related to sugar compounds accounted for a large proportion of the whole community, and suggested that sugar compounds were driven by a few taxa. However,

since the types and amounts of sugar species produced by microorganisms are affected by microbial species and external environmental conditions such as carbon sources, temperature and humidity (Hrynkiewicz et al., 2010), the corresponding relations between microorganisms and sugar compounds vary with location. The presented results in this study indicate that sugar compounds were significantly associated with the fungal community structure, but not with the bacterial community structure. At the taxon level, only six minor fungal genera (*Peniophora*, *Sporobolomyces*, *Hydnophlebia*, *Hannaella*, *Lectera*

and *Cyphellophora*) were found to be associated with three sugar alcohols (mannitol, arabitol and erythritol) and trehalose, while no bacterial taxon was significantly related to sugar molecules. In addition, the above three sugar alcohols, galactosan, levoglucosan, and trehalose showed a significant positive correlation ($p < 0.05$) with fungal OTU abundance (**Table S4**). These results suggested that sugar compounds in rural aerosols had a strong association with overall fungal community structure (based on the Mantel test and correlation analysis) and especially a few fungal taxa (based on co-occurrence analysis), while

bacteria have a weak effect on sugar compounds possibly due to their limited mass.

## 4 Conclusions

In this study, the airborne microbial taxonomy and the compositions of sugar molecules in atmospheric aerosols collected in rural Gucheng, North China Plain were obtained. Results confirm that fungal communities have more pronounced diurnal differences than bacteria. Airborne fungi mainly originated from plant leaves, while the sources of airborne bacteria were

mostly unresolved, which may be human-related and/or long-distance transport sources. Among sugar compounds, sucrose dominated in the daytime samples ($27.2 \pm 12.7\%$), while mannitol dominated at night ($31.1 \pm 5.9\%$). From the perspective of the overall microbial community, most sugar molecules were strongly associated with the fungal community structure, while the bacteria community structure was weakly related to sucrose only. At the microbial taxon level, only a few saprophytic fungal taxa were closely associated with sugar compounds. The results suggest that the entire fungal community may largely

contribute to sugar compounds, while bacterial community contributed little to them. The presented results in this study will help to understand the contribution of bioaerosols to organic carbon in atmospheric aerosols and build links between airborne microorganisms and aerosol chemistry. Importantly, it should be noted that the uncertainties in estimating the concentrations and emission fluxes of bioaerosols using certain sugar compounds need to be seriously considered. For example, arabitol, usually used as a fungal spore tracer, showed no significant correlation with fungi in this study. More likely, different seasons,

climates and locations may lead to differences in the microbial community structure as well as corresponding sugar composition and metabolites.

Moreover, the airborne microbiota is highly complex, and biomolecules such as carbohydrates, proteins and lipids are major chemical compositions and metabolites of airborne microorganisms. In this study, the measured carbohydrate species are
limited to small molecule compounds. In the future, it is necessary to detect more carbohydrates and other compounds in
aerosols from different typical regions by using high-resolution mass spectrometry to further study the relationship between
microorganisms and atmospheric aerosol chemistry.

## Data availability

The sequence data is available in the NCBI Sequence Read Archive under accession numbers SUB12879673 and
SUB12879615.

**Author contributions**

MTN analyzed data and prepared all figures. SH and WW conducted the observation. SH, YJW, RJ, QZ, ZHW and DHZ
performed experiments. MTN and WH wrote the manuscript. WYX, LBW, JJD, FXS and PQF revised the original manuscript
draft. WH and PQF supervised the project. All authors were involved in helpful discussions and contributed to the manuscript.

## Competing interests

The contact authors have declared that none of the authors has any competing interests.

## Disclaimer

Publisher's note: Copernicus Publications remains neutral about jurisdictional claims in published maps and institutional
affiliations

## Acknowledgements

Thanks for the help of the staffs at the Gucheng Ecological and Agrometeorological Experiment Station, the Chinese Academy
of Meteorological Sciences during the observation.

## Financial support

This work was supported by the National Natural Science Foundation of China (Nos. 41977183, 42130513).



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



**Figure 1.** Distribution and comparison of (a) bacterial and (b) fungal diversity and richness in the daytime and nighttime samples. The dots in the violin shapes represent all data points. Stars represent the average value of all data. *, $p < 0.05$; **, $p < 0.01$; ns, no significant difference.



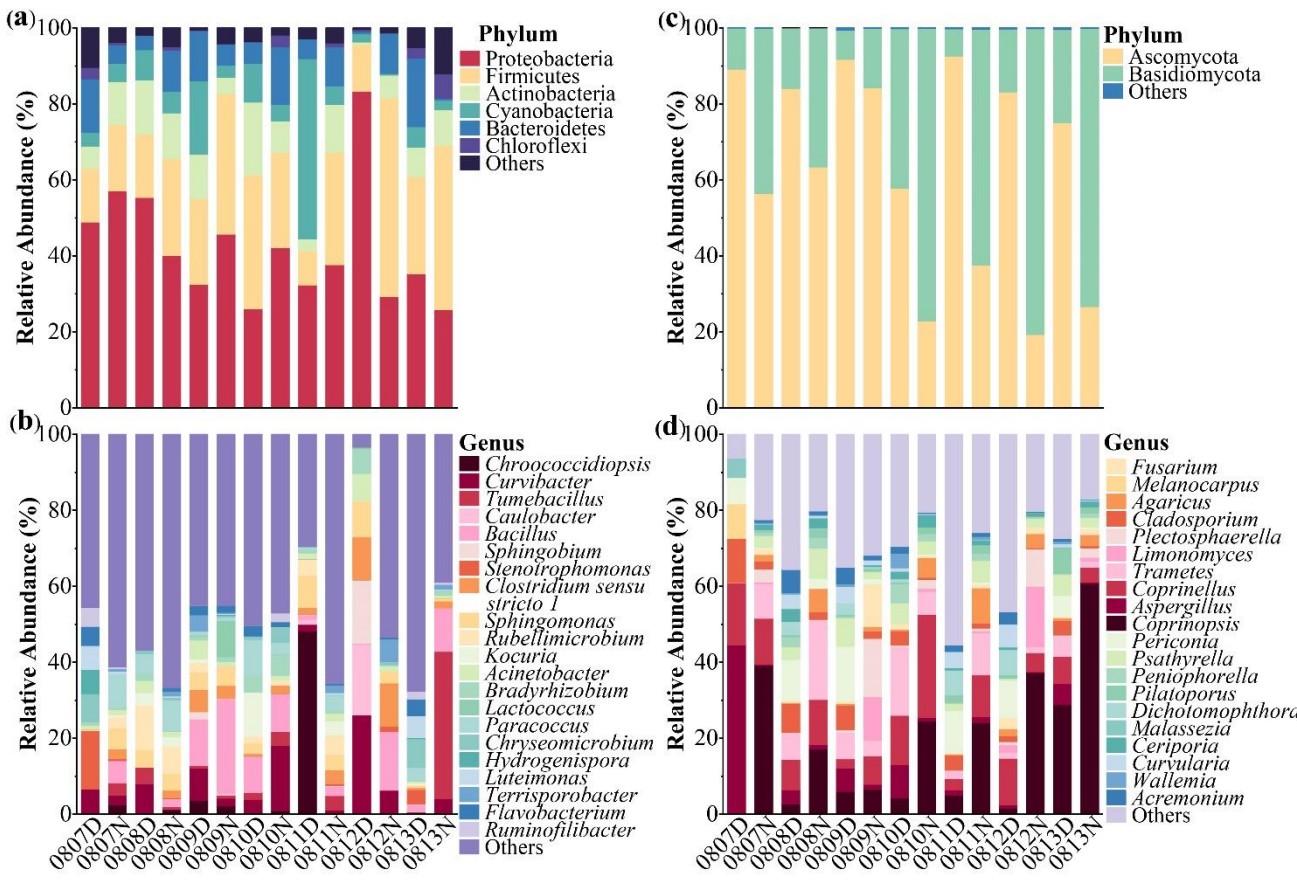


**Figure 2.** Relative abundances of dominant bacterial (a−b) and fungal (c−d) aerobiota at the phylum (upper panel) and genus (bottom panel) levels. D and N in the sample IDs mean daytime and nighttime, respectively.





**Figure 3.** Significant differences between the relative abundances of (a) bacterial and (b) fungal phyla, orders and genera in diurnal samples
analyzed by one-way ANOVA analysis.





**Figure 4.** (a) Contribution of potential sources to airborne bacteria and fungi estimated by SourceTracker analysis. (b) The relationship between bacterial or fungal communities and environmental factors based on redundancy analysis (RDA). T, air temperature; RH, relative humidity; WS, wind speed.





**Figure 5.** (a) Concentrations of sugars and (b) the contributions to organic carbon (OC) from plant-debris OC, fungal spore OC and biomass-burning OC to aerosol OC (%) in rural aerosols at Gucheng, China.




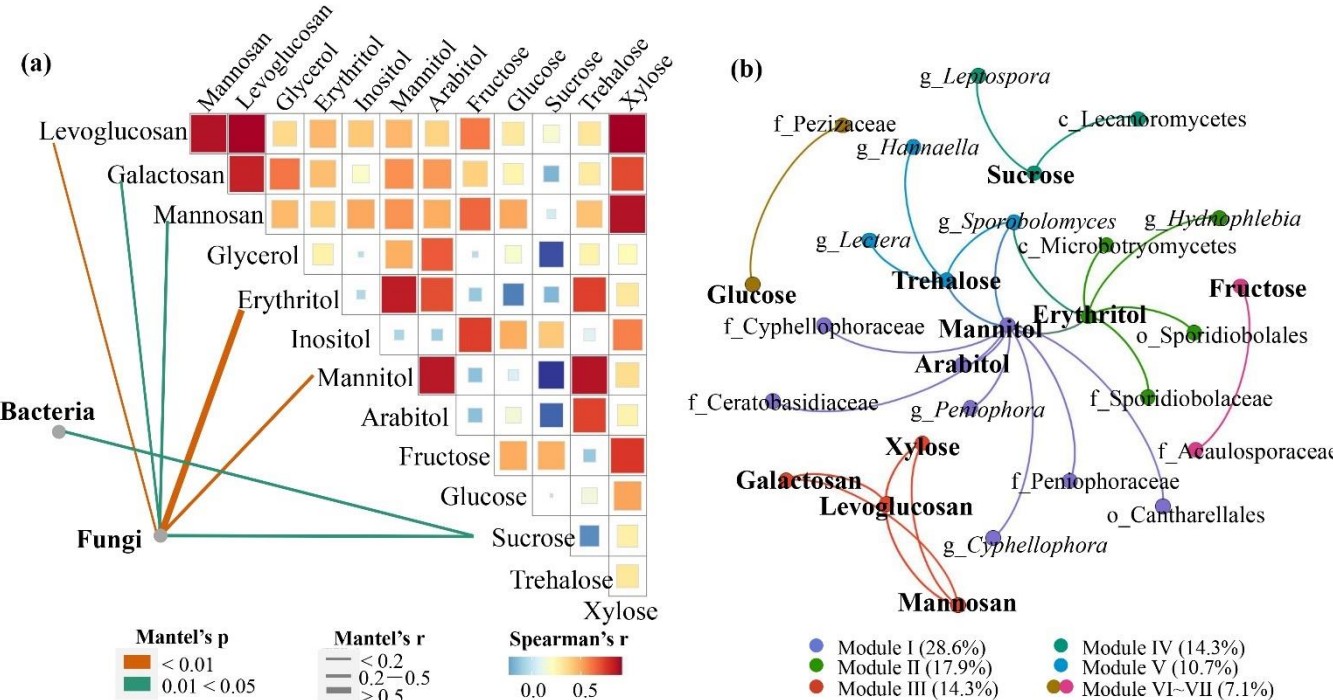

**Figure 6.** (a) Correlations of the microbial community structures with sugar compounds based on Bray–Curtis distance. Edge width corresponds to Mantel's r value, and the edge color denotes statistical significance. Pairwise correlations of sugar compounds with a heatmap are calculated by Spearman's correlation analysis. (b) Spearman's correlation-based co-occurrence analysis between genera and sugar components. The nodes are colored according to different types of modularity classes.