# Peer review of "Characteristics of bacterial and fungal communities and their associations with sugar compounds in atmospheric aerosols at a rural site in North China"

_Biogeosciences, 2023_

## Referee Comment (RC3)

The manuscript reports a study on the connection between microbial communities and sugar compounds in atmospheric aerosols. The offline filter-based samples of total suspended particles (TSP) were collected in a one-week campaign. DNA sequencing data and sugar compounds were obtained to investigate their relationships. The study gives an attempt to capture and compare bioaerosol characteristics with different techniques. The manuscript fits the scope of the journal and could be considered for publication after revision. My concerns and suggestions are shown as follows:

(1) It is a good idea to create a linkage between microbial communities and organic tracers. It should always be aware that these connections can be sensitively affected by the sources of bioaerosols as well as environmental factors. Therefore, the nearby sources and other related activities should be mentioned in the manuscript and their influence should be carefully considered. A well-defined sampling area/site and the period should larging reduce the unexpected interference.

(2) Source Tracker method was used to distinguish the sources. The details of this method should be provided especially giving more information such as how the source data were combined (soil, leaves …).

(3) The shown RDA analysis for bacteria seems less confident than that for fungi. Can you explain why this happens? Due to the small dataset or just the more complicated sources and influencing factors of bacteria in ambient environment? The Source Tracker method did not work very well for bacteria with large percentage of unresolved sources as shown in Line 214. I expect that it should be a quite common phenomenon in ambient air which has been suggested by some other studies. But still, a discussion is necessary here.

(4) Section 3.2 can be shortened. Some discussions/claims are not necessarily true due to the small dataset presented here. In another word, it is not your job to investigate the influence other possible factors. I would suggest putting more efforts on the diurnal difference of fungal aerosols which has been clearly demonstrated by the community structure and sugar tracers. It is very interesting and needs more data analysis to generate in-depth generalizations.

(5) In general, I would suggest shortening the part of results which only presents data and describes the variation. More importantly, Section 3.4.2 should be polished with the emphasis on the used method and details of interaction. I do not fully understand how the interaction happened. How about the sensitivity of the interaction? For example, did the nighttime processes give more weight on the interaction?

---

## Author Response (AR1)

**Response to Reviewers**

Editor of *Biogeosciences*

Dear Prof. Yu Luo,

We gratefully thank the editor and the reviewers for their time making their constructive remarks and useful suggestions, which has significantly raised the quality of the manuscript and has enable us to improve the manuscript. Each suggested revision and comment brought forward by the reviewers was accurately incorporated and considered. Below the comments of the reviewers are responded point by point and the revisions are indicated in blue color in the marked manuscript.

Sincerely,

HU Wei (on behalf of coauthors)

**Reviewer #1**

The authors investigated airborne bacterial and fungal communities and their associations with sugar compounds at a typical rural site, Dingxing County, in the North China Plain. Their community structure, source, diurnal variation, possible factors, and contribution to sugar compounds were addressed in this manuscript. The results are conducive to deepening the understanding of bioaerosol and those potential influence on the enhanced sugar compounds at nighttime. This is a valuable piece of work also with a reasonable overall structure. I recommend that it can be accepted for Minor Revision.

**Response:**

We greatly appreciate the reviewer for the professional comments and suggestions. According to the valuable suggestions, we have made extensive corrections to the original manuscript. The corrections are marked in blue in the revised manuscript. Please refer to the point-to-point responses to the comments below.

**Comment 1**

For the SourceTracker, soils, plant leaves and seawater were selected as the source tracking reference databases. But the sources of bacterial community were mostly unresolved (about 98.9%). I wondered whether the database was precise or not? Is there any seawater around the sampling site? Did the nearby river water contribute more to the bacterial community?

**Response:**

Previous studies reported that local sources played leading roles in shaping microbial communities (Zhai et al., 2018; Bowers et al., 2011). Hence, we only consider the contributions of local natural sources to microbial communities in this study. To ensure as much accuracy as possible, the leaf and soil samples we chose were sampled near the atmospheric sampling site. However, due to the small dataset, there is still a large uncertainty. The closest sea area to the sampling site is about 178 km away. There are rivers nearby (7.7 km), and we did not find sequencing data of nearby river waters in the NCBI database. The large unresolved sources of this study may be partially corroborated by the result of previous studies that the bacteria were more likely to be of anthropogenic origin (Jiang et al., 2022; Zhao et al., 2022).

For the large uncertainty of the SourceTracker results, we explained the possible reasons in Page 8, Line 237-242: *"only the contribution of natural sources including terrestrial plants, soils and ocean to the airborne microbial community was considered in this study, which may partly explain the high unresolved sources of bacteria. It is also noticeable that in this study only local soil and vegetation samples were selected as the source samples, and the impact of long-distance transport was not considered. Therefore, unresolved sources may also come from long-range transported air masses (Gat et al., 2017) and other local sources, e.g., waters, other vegetations and human-related sources."*

Page 11, Line 332-335: *"It was also found that bacterial community structure may be closely related to sucrose-traced plant origins, although the SourceTracker analysis results did not show the plant origin of bacteria. This inconsistency may be caused by single plant (maize) source samples and/or the negligence of long-distance transport in this study."*

The detailed description of SourceTracker was added as follows:

Page 5, line 141-148: *"SourceTracker is a Bayesian approach to identifying sources and estimating source proportions for microbial surveys (Knights et al., 2011). The source samples used as source tracking reference databases in this study were soils, plant leaves and seawater. Maize leaves and soil samples were collected around the observation site and the samples were subjected to high-throughput sequencing as described in Section 2.2 to obtain the gene sequences, and the marine microbial sequences were obtained from the NCBI SRA database (https://www.ncbi.nlm.nih.gov/sra/, Table S1). The sequences of all source and sink samples were processed together by QIIME 2, and were assigned to OTUs based on the similarity of sequences ≥ 97%. The output data were performed for source tracking analysis based on "SourceTracker2" package in R (https://github.com/caporaso-lab/sourcetracker2)."*

Bowers, R. M., McLetchie, S., Knight, R., and Fierer, N.: Spatial variability in airborne bacterial communities across land-use types and their relationship to the bacterial communities of potential source environments, ISME J., 5, 601-612, https://doi.org/10.1038/ismej.2010.167, 2011.

Jiang, X. Q., Wang, C. H., Guo, J. Y., Hou, J. H., Guo, X., Zhang, H. Y., Tan, J., Li, M., Li, X., and Zhu, H. Q.: Global meta-analysis of airborne bacterial communities and associations with anthropogenic activities, Environ. Sci. Technol., 56, 9891-9902, https://doi.org/10.1021/acs.est.1c07923, 2022.

*Gat, D., Mazar, Y., Cytryn, E., and Rudich, Y.: Origin-dependent variations in the atmospheric microbiome community in Eastern Mediterranean dust storms, Environ. Sci. Technol., 51, 6709-6718, https://doi.org/10.1021/acs.est.7b00362, 2017.*

*Knights, D., Kuczynski, J., Charlson, E. S., Zaneveld, J., Mozer, M. C., Collman, R. G., Bushman, F. D., Knight, R., and Kelley, S. T.: Bayesian community-wide culture-independent microbial source tracking, Nat. Methods, 8, 761-763, https://doi.org/10.1038/nmeth.1650, 2011.*

*Zhai, Y., Li, X., Wang, T., Wang, B., Li, C., and Zeng, G.: A review on airborne microorganisms in particulate matters: Composition, characteristics and influence factors, Environ. Int., 113, 74-90, https://doi.org/10.1016/j.envint.2018.01.007, 2018.*

*Zhao, J., Jin, L., Wu, D., Xie, J., Li, J., Fu, X., Cong, Z., Fu, P., Zhang, Y., Luo, X., Feng, X., Zhang, G., Tiedje, J. M., and Li, X.: Global airborne bacterial community—Interactions with Earth's microbiomes and anthropogenic activities, Proc. Natl. Acad. Sci. U. S. A., 119, e2204465119, https://doi.org/10.1073/pnas.2204465119, 2022.*

**Comment 2**

It seems that the environmental factors did not explain adequate variance of bacterial community also by the RDA analysis. The cumulative explanatory variable of RDA1 and RDA2 axis were lower than 60%, hence did the RDA analysis have statistical significance or more factors should be included.

**Response:**

Environmental factors did not explain the bacterial community changes well in this study. We favor the view that other ecological processes played an important role in the construction of bacterial communities. We applied inferring community assembly mechanisms by phylogenetic-bin-based null model analysis (iCAMP) to unravel the drivers controlling community assembly (Ning et al., 2020) and found that deterministic processes (including environmental filtering and biological interactions) and stochastic processes (birth/death, speciation/extinction, and immigration) played nearly equal roles (53.7% and 46.3%) in shaping bacterial communities. The corresponding information was revised as follows:

Page 5, line 154-157: "*Inferring community assembly mechanisms by phylogenetic-bin-based null model (iCAMP) is a general framework to quantitatively estimate community assembly*

*mechanisms (Ning et al., 2020; Chen et al., 2022). The assembly mechanisms of airborne microbes at Gucheng site were investigated by using the iCAMP package in R."*

Page 8-9, line 253-258: "*It should be noted that environmental factors explained only 43.4% of the variation in bacterial communities, and it is hypothesized that ecological processes other than environmental filtration had impacts on shaping bacterial communities. Here, iCAMP analysis showed that the relative contributions of deterministic processes (environmental filtration and microbial interactions) and stochastic processes (birth/death, speciation/extinction, and immigration) to shaping bacterial communities were 53.7% and 46.3%, respectively, demonstrating the importance of other ecological processes on bacterial communities.*"

*Chen, D., Hou, H., Zhou, S., Zhang, S., Liu, D., Pang, Z., Hu, J., Xue, K., Du, J., Cui, X., Wang, Y., and Che, R.: Soil diazotrophic abundance, diversity, and community assembly mechanisms significantly differ between glacier riparian wetlands and their adjacent alpine meadows, Front. Microbiol., 13, https://doi.org/10.3389/fmicb.2022.1063027, 2022.*

*Ning, D., Yuan, M., Wu, L., Zhang, Y., Guo, X., Zhou, X., Yang, Y., Arkin, A. P., Firestone, M. K., and Zhou, J.: A quantitative framework reveals ecological drivers of grassland microbial community assembly in response to warming, Nature Communications, 11, 4717, https://doi.org/10.1038/s41467-020-18560-z, 2020.*

**Comment 3**

The results showed that nighttime fungal spore emitted more OC than that in daytime ($p < 0.01$, Fig 5b), however daytime samples were more affected by OC than the nighttime samples based on the redundancy analysis (Fig 4b). How to evaluate the different phenomenon?

**Response:**

Thank the reviewer for the valuable comment. The fungal-spore OC shown in Fig. 5b was estimated by the concentrations of mannitol based on Bauer et al. (2008), and the fungal-spore OC accounted for 3.6–19.7% (average: 7.7%) of total OC. Whereas, the parameter OC introduced into the RDA analysis (Fig 4b) was the total concentration of OC. Therefore, this phenomenon may be due to the fact that non-fungal-spore OC played role in influencing the daytime fungal community. In addition to the possible microbial origin of OC, OC may be an available nutrient for microbial development (Vaïtilingom et al., 2013;Wei et al., 2019).

Vaïtilingom, M., Deguillaume, L., Vinatier, V., Sancelme, M., Amato, P., Chaumerliac, N., and Delort, A. M.: Potential impact of microbial activity on the oxidant capacity and organic carbon budget in clouds, Proc. Natl. Acad. Sci. USA, 110, 559-564, https://doi.org/10.1073/pnas.1205743110, 2013.

Wei, M., Xu, C., Xu, X., Zhu, C., Li, J., and Lv, G.: Size distribution of bioaerosols from biomass burning emissions: Characteristics of bacterial and fungal communities in submicron (PM1.0) and fine (PM2.5) particles, Ecotoxicol. Environ. Saf., 171, 37-46, https://doi.org/10.1016/j.ecoenv.2018.12.026, 2019.

Thank you very much for your comments and suggestions. Your any further comments and suggestions are appreciated.

**Reviewer #2**

The authors investigated the association of sugar molecules and the microbial community at taxonomic levels in atmospheric aerosols and found that sugar compounds exhibited significant associations with fungal community structure than bacterial community structure. And this strong associations between fungal community and sugar compounds in rural aerosols may due to the entire fungal community rather than specific fungal taxa. Overall, this study was well-designed, interesting and complete. The results observed could add substantially to the literature, provide a reference for further studies exploring the contribution of bioaerosols to organic carbon in atmospheric aerosols and build links between airborne microorganisms and aerosol chemistry. I would suggest the publication of this manuscript after addressing the following comments.

**Response:**

Thank the reviewer for the valuable and professional comments. The corrections are marked in blue in the revised manuscript. Please refer to the point-to-point responses to the comments below.

**Comment 1**

Since the samples were collected in the summer of 2020 (7−13 August). Did the high temperature affect the association of sugar molecules and the microbial community at taxonomic levels in atmospheric aerosols?

**Response:**

Yes, it's quite possible. Microbial communities and sugar molecules were significantly affected by environmental factors (Zhai et al., 2018;Fu et al., 2012), and the types and amounts of sugar species produced by microorganisms are affected by microbial species and external environmental conditions such as carbon sources, temperature and humidity (Hrynkiewicz et al., 2010). Therefore, we consider that the corresponding relations between microorganisms and sugar compounds vary with the external environmental factors, e.g., temperature, relative humidity and geographical location. To make it clearer, the following sentence was revised as follows:

Page 12, line 364-373: *"In addition, it should be noted that the taxa associated with sugar compounds found in this study were not the same as those found in a survey conducted by Samaké et al. (2020) at a rural site in France, the only study up-to-date investigating the association between*

*microorganisms and sugar compounds. The types and amounts of sugar species produced by microorganisms are affected by microbial species and external environmental conditions such as carbon sources, temperature and humidity (Hrynkiewicz et al., 2010). Here, we performed network construction separately for daytime and nighttime samples. Results showed that the sugar molecules had more associations with microbial taxa at night, particularly fungal taxa (Fig. S7), demonstrating the importance of environmental factors on the relationship between sugar compounds and microbial taxa. More fungal spores could be released at night due to high temperature and high relative humidity (Elbert et al., 2007;Hummel et al., 2015) and the proportions of some fungal genera (e.g., Agaricus) enhanced at night (Table S3), which was a possible explanation for the greater associations between sugar molecules and microbial taxa at night."*

The related analysis was added in Figure S7 as follows:

[Figure]

**Figure S7.** Co-occurrence networks between microbial taxa and sugar components (bold font) were constructed based on Spearman's correlation analysis in the (a) daytime and (b) nighttime samples. The nodes are colored according to different types of modularity classes.

*Elbert, W., Taylor, P. E., Andreae, M. O., and PöSchl, U.: Contribution of fungi to primary biogenic aerosols in the atmosphere: Wet and dry discharged spores, carbohydrates, and inorganic ions, Atmos. Chem. Phys., 7, 4569-4588, 2007.*

*Fu, P., Kawamura, K., Kobayashi, M., and Simoneit, B. R. T.: Seasonal variations of sugars in atmospheric particulate matter from Gosan, Jeju Island: Significant contributions of airborne pollen and Asian dust in spring, Atmos. Environ., 55, 234-239, https://doi.org/10.1016/j.atmosenv.2012.02.061, 2012.*

*Hrynkiewicz, K., Baum, C., and Leinweber, P.: Density, metabolic activity, and identity of cultivable rhizosphere bacteria on Salix viminalis in disturbed arable and landfill soils, J. Plant Nutr. Soil Sci., 173, 747-756, https://doi.org/10.1002/jpln.200900286, 2010.*

*Hummel, M., Hoose, C., Gallagher, M., Healy, D. A., Huffman, J. A., O'Connor, D., Pöschl, U., Pöhlker, C., Robinson, N. H., Schnaiter, M., Sodeau, J. R., Stengel, M., Toprak, E., and Vogel, H.: Regional-scale simulations of fungal spore aerosols using an emission parameterization adapted to local measurements of fluorescent biological aerosol particles, Atmos. Chem. Phys., 15, 6127-6146, https://doi.org/10.5194/acp-15-6127-2015, 2015.*

*Zhai, Y., Li, X., Wang, T., Wang, B., Li, C., and Zeng, G.: A review on airborne microorganisms in particulate matters: Composition, characteristics and influence factors, Environ. Int., 113, 74-90, https://doi.org/10.1016/j.envint.2018.01.007, 2018.*

**Comment 2**

Why the authors chose the TSP samples? If the authors wanted to investigate the contribution of bioaerosols to organic carbon in atmospheric aerosols, it's better to collect PM2.5 or PM10 samples.

**Response:**

Although the particle sizes of microorganisms are usually smaller than 10 μm, microorganisms tend to attach to the particulate matter for airborne transport (Fröhlich-Nowoisky et al., 2016). Focusing on sugar compounds and microorganisms in TSP samples may offer a broader perspective on the relationship between sugar compounds and microorganisms.

"*Considering the broad size range of airborne microorganisms and their attachment to other coarse particles (Fröhlich-Nowoisky et al., 2016)*" was added in Line 88.

*Fröhlich-Nowoisky, J., Kampf, C. J., Weber, B., Huffman, J. A., Pöhlker, C., Andreae, M. O., Lang-Yona, N., Burrows, S. M., Gunthe, S. S., and Elbert, W.: Bioaerosols in the Earth system: Climate, health, and ecosystem interactions, Atmos. Res., 182, 346-376, 2016.*

**Comment 3**

The microbial community may be affected by air quality parameters such as PM$_{2.5}$, PM$_{10}$, NO$_2$, O$_3$, SO$_2$ and CO. The sugar compounds may be also affected by these pollution factors. I suggest the authors add some analysis about these parameters (also the air temperature).

**Response:**

The content about the relationship between sugar compounds and environmental factors was added as follows:

Page 9, line 272-273: *"Moreover, the concentrations of anhyrosugars were significantly affected by temperature (Fig. S6), which may also be responsible for the diurnal differences in anhyrosugars."*

Page 10, line 304-306: *"Meanwhile, the temperature significantly affected the sugar alcohols (Fig. S6). Oduber et al. (2021) also reported high correlations between sugar alcohols and air temperature. This phenomenon may be corroborated by the finding that the activities of fungal spores were regulated by temperature (Zhu et al., 2015)."*

The related analysis was added in Figure S6 and revised in Table S5 as follows:

[Figure]

**Figure S6.** Correlations of sugar compounds with environmental factors based on Bray–Curtis distance. Edge width corresponds to Mantel's r value, and the edge color denotes statistical significance. Pairwise correlations of environmental factors with a heatmap are calculated by Spearman's correlation analysis.

**Table S5.** Spearman correlation of sugar components and environmental or fungal α-diversity indices.

| Factors | Anhyrosugars | | | Sugar alcohols | | | | | Primary saccharides | | | | |
|---|---|---|---|---|---|---|---|---|---|---|---|---|---|
| | Galactosan | Mannosan | Levoglucosan | Arabitol | Mannitol | Erythritol | Inositol | Glycerol | Glucose | Sucrose | Fructose | Xylose | Trehalose |
| Galactosan | 1 | | | | | | | | | | | | |
| Mannosan | 0.833** | 1 | | | | | | | | | | | |
| Levoglucosa | 0.881** | 0.916** | 1 | | | | | | | | | | |
| Arabitol | 0.547* | 0.503 | 0.398 | 1 | | | | | | | | | |
| Mannitol | 0.569* | 0.560* | 0.477 | 0.864** | 1 | | | | | | | | |
| Erythritol | 0.459 | 0.411 | 0.477 | 0.719** | 0.851** | 1 | | | | | | | |
| Inositol | 0.2 | 0.516 | 0.433 | -0.086 | -0.064 | -0.051 | 1 | | | | | | |
| Glycerol | 0.626* | 0.473 | 0.38 | 0.688** | 0.499 | 0.284 | -0.02 | 1 | | | | | |
| Glucose | 0.27 | 0.516 | 0.323 | 0.16 | 0.073 | -0.305 | 0.503 | 0.187 | 1 | | | | |
| Sucrose | -0.169 | 0.051 | 0.182 | -0.371 | -0.473 | -0.16 | 0.429 | -0.42 | -0.007 | 1 | | | |
| Fructose | 0.407 | 0.662** | 0.618* | -0.134 | -0.13 | -0.108 | 0.754** | -0.024 | 0.508 | 0.495 | 1 | | |
| Xylose | 0.727** | 0.881** | 0.921** | 0.279 | 0.367 | 0.327 | 0.604* | 0.244 | 0.525 | 0.292 | 0.771** | 1 | |
| Trehalose | 0.31 | 0.446 | 0.336 | 0.745** | 0.877** | 0.749** | 0.09 | 0.275 | 0.156 | -0.275 | -0.103 | 0.332 | 1 |
| OC | 0.653* | 0.855** | 0.881** | 0.429 | 0.385 | 0.433 | 0.411 | 0.288 | 0.341 | 0.411 | 0.622* | 0.829** | 0.345 |
| $Ca^{2+}$ | 0.543* | 0.534* | 0.653* | 0.556* | 0.415 | 0.547* | 0.218 | 0.367 | 0.055 | 0.358 | 0.138 | 0.486 | 0.415 |
| $Cl^{-}$ | 0.829** | 0.811** | 0.749** | 0.798** | 0.719** | 0.547* | 0.116 | 0.789** | 0.292 | -0.244 | 0.191 | 0.582* | 0.49 |
| $SO_4^{2-}$ | 0.446 | 0.705** | 0.560* | 0.187 | 0.125 | 0.055 | 0.587* | 0.178 | 0.508 | 0.42 | 0.829** | 0.670** | 0.156 |
| Temperature | -0.196 | 0.033 | 0.046 | -0.644* | -0.670** | -0.604* | 0.411 | -0.464 | 0.31 | 0.679** | 0.631* | 0.235 | -0.604* |
| RH | 0.407 | 0.244 | 0.181 | 0.608* | 0.806** | 0.670** | -0.297 | 0.445 | -0.207 | -0.705** | -0.313 | 0.053 | 0.698** |
| Wind speed | -0.613* | -0.49 | -0.675** | -0.407 | -0.495 | -0.662** | -0.029 | -0.244 | 0.213 | -0.046 | -0.015 | -0.415 | -0.376 |
| $PM_{2.5}$ | 0.152 | 0.187 | -0.007 | 0.117 | 0.361 | 0.442 | 0.337 | 0.18 | 0.09 | -0.233 | 0.037 | -0.004 | 0.429 |
| $PM_{10}$ | 0.521 | 0.732** | 0.609* | 0.635* | 0.53 | 0.336 | 0.288 | 0.389 | 0.640* | 0.174 | 0.398 | 0.662** | 0.534* |
| $SO_2$ | 0.031 | 0.277 | 0.077 | 0.29 | 0.242 | 0.141 | 0.031 | -0.024 | 0.099 | -0.042 | 0.064 | -0.02 | -0.053 |
| $NO_2$ | -0.126 | 0.042 | -0.051 | -0.267 | -0.294 | -0.373 | -0.289 | 0.139 | 0.104 | 0.241 | 0.435 | -0.038 | -0.448 |
| $O_3$ | 0.468 | 0.53 | 0.464 | 0.459 | 0.899** | 0.899** | 0.815** | -0.011 | 0.099 | -0.16 | -0.138 | 0.354 | 0.859** |
| OTU | 0.552* | 0.503 | 0.534* | 0.679** | 0.837** | 0.785** | 0.051 | 0.336 | -0.011 | -0.292 | -0.086 | 0.433 | 0.851** |
| Shannon | 0.415 | 0.424 | 0.257 | 0.402 | 0.516 | 0.095 | 0.125 | 0.292 | 0.604* | -0.635* | 0.015 | 0.279 | 0.525 |

Note: *$p < 0.01$; **$p < 0.05$.

*Zhu, C., Kawamura, K., and Kunwar, B.: Organic tracers of primary biological aerosol particles at subtropical Okinawa Island in the western North Pacific Rim, J. Geophys. Res. Atmos., 120, 5504-5523, https://doi.org/10.1002/2015JD023611, 2015.*

*Oduber, F., Calvo, A. I., Castro, A., Alves, C., Blanco-Alegre, C., Fernández-González, D., Barata, J., Calzolai, G., Nava, S., Lucarelli, F., Nunes, T., Rodríguez, A., Vega-Maray, A. M., Valencia-Barrera, R. M., and Fraile, R.: One-year study of airborne sugar compounds: Cross-interpretation with other chemical species and meteorological conditions, Atmos. Res., 251, 105417, https://doi.org/10.1016/j.atmosres.2020.105417, 2021.*

**Comment 4**

The authors used the filters to collect the sugar samples. Did the author test the recovery efficiencies of these sugar compounds from the filter?

**Response:**

Yes, we tested the recovery efficiencies of these sugar compounds and the recoveries ranged from 80% to 120%. To make it clearer, the following sentence was added:

Page 5, line 131-132: *"The recoveries of sugar compounds ranged from 80% to 120%. Reproducibility of the method was larger than 90%."*

Thank you very much for your comments and suggestions. Your any further comments and suggestions are appreciated.

**Reviewer #3**

The manuscript reports a study on the connection between microbial communities and sugar compounds in atmospheric aerosols. The offline filter-based samples of total suspended particles (TSP) were collected in a one-week campaign. DNA sequencing data and sugar compounds were obtained to investigate their relationships. The study gives an attempt to capture and compare bioaerosol characteristics with different techniques. The manuscript fits the scope of the journal and could be considered for publication after revision.

**Response:**

Thank the reviewer for the detailed comments and suggestions on the content of the manuscript. We have shortened the section describing the results and revised the whole text as suggested. All the contents revised accordingly were highlighted with blue color. Please see the point-to-point responses to the detailed comments below.

**Comment 1**

It is a good idea to create a linkage between microbial communities and organic tracers. It should always be aware that these connections can be sensitively affected by the sources of bioaerosols as well as environmental factors. Therefore, the nearby sources and other related activities should be mentioned in the manuscript and their influence should be carefully considered. A well-defined sampling area/site and the period should reduce the unexpected interference

**Response:**

Thank the reviewer for the valuable comment. A more detailed description of the sampling site and the sampling period was added as follows:

Page 3, line 83-87: *"The observation site is located in Dingxing County, Hebei Province, which is about 100 km and 128 km away from two nearby megacities Beijing and Tianjin, respectively (Fig. S1). The site is located in a rural area in the north of the North China Plain. It is a high-yield agricultural area with maize as the main summer crop, about 8 km from the nearest river and 178 km from the closest ocean. The site is about 900 m away from the nearest railway and 2.7 km from the*

*expressway. There was no intense agricultural activity observed around the site during the observation period."*

**Comment 2**

SourceTracker method was used to distinguish the sources. The details of this method should be provided especially giving more information such as how the source data were combined (soil, leaves …).

**Response:**

The detailed description of SourceTracker was added as follows:

Page 5, line 141-148: *"SourceTracker is a Bayesian approach to identifying sources and estimating source proportions for microbial surveys (Knights et al., 2011). The source samples used as source tracking reference databases in this study were soils, plant leaves and seawater. Maize leaves and soil samples were collected around the observation site and the samples were subjected to high-throughput sequencing as described in Section 2.2 to obtain the gene sequences, and the marine microbial sequences were obtained from the NCBI SRA database (https://www.ncbi.nlm.nih.gov/sra/, Table S1). The sequences of all source and sink samples were processed together by QIIME 2, and were assigned to OTUs based on the similarity of sequences $\geq$ 97%. The output data were performed for source tracking analysis based on "SourceTracker2" package in R (https://github.com/caporaso-lab/sourcetracker2)."*

*Knights, D., Kuczynski, J., Charlson, E. S., Zaneveld, J., Mozer, M. C., Collman, R. G., Bushman, F. D., Knight, R., and Kelley, S. T.: Bayesian community-wide culture-independent microbial source tracking, Nat. Methods, 8, 761-763, https://doi.org/10.1038/nmeth.1650, 2011.*

**Comment 3**

The shown RDA analysis for bacteria seems less confident than that for fungi. Can you explain why this happens? Due to the small dataset or just the more complicated sources and influencing factors of bacteria in ambient environment? The SourceTracker method did not work very well for bacteria with large percentage of

unresolved sources as shown in Line 214. I expect that it should be a quite common phenomenon in ambient air which has been suggested by some other studies. But still, a discussion is necessary here.

**Response:**

We consider that other ecological processes may play important roles in the shaping bacterial communities. We applied inferring community assembly mechanisms by phylogenetic-bin-based null model analysis (iCAMP) to unravel the drivers controlling community assembly (Ning et al., 2020) and found that deterministic processes (including environmental filtering and biological interactions) and stochastic processes (birth/death, speciation/extinction, and immigration) played nearly equal roles (53.7% and 46.3%) in shaping bacterial communities. Some studies have indeed found that natural sources contributed less to the airborne bacterial community (Jiang et al., 2022;Zhao et al., 2022). In addition, small datasets may also be responsible for large percentage of unresolved sources. The corresponding information was revised as follows:

Page 5, line 154-157: *"Inferring community assembly mechanisms by phylogenetic-bin-based null model (iCAMP) is a general framework to quantitatively estimate community assembly mechanisms (Ning et al., 2020; Chen et al., 2022). The assembly mechanisms of airborne microbes at Gucheng site were investigated by using the iCAMP package in R."*

Page 8, line 234-242: *"Zhao et al. (2022) found that human-related sources contributed greater to the airborne bacteria than other sources in urban areas. Bacterial communities are more likely to be influenced by human body surface flora than other environments (Jiang et al., 2022). Bacterial community over the Eastern Mediterranean originated likewise mainly from anthropogenic sources (Mazar et al., 2016). However, only the contribution of natural sources including terrestrial plants, soils and ocean to the airborne microbial community was considered in this study, which may partly explain the high unresolved sources of bacteria. It is also noticeable that in this study only local soil and vegetation samples were selected as the source samples, and the impact of long-distance transport was not considered. Therefore, unresolved sources may also come from long-range transported air masses (Gat et al.,*

*2017) and other local sources, e.g., waters, other vegetations and human-related sources."*

Page 8-9, line 253-258: *"It should be noted that environmental factors explained only 43.4% of the variation in bacterial communities, and it is hypothesized that ecological processes other than environmental filtration had impacts on shaping bacterial communities. Here, iCAMP analysis showed that the relative contributions of deterministic processes (environmental filtration and microbial interactions) and stochastic processes (birth/death, speciation/extinction, and immigration) to shaping bacterial communities were 53.7% and 46.3%, respectively, demonstrating the importance of other ecological processes on bacterial communities."*

Page 11, Line 332-335: *"It was also found that bacterial community structure may be closely related to sucrose-traced plant origins, although the SourceTracker analysis results did not show the plant origin of bacteria. This inconsistency may be caused by single plant (maize) source samples and/or the negligence of long-distance transport in this study."*

Chen, D., Hou, H., Zhou, S., Zhang, S., Liu, D., Pang, Z., Hu, J., Xue, K., Du, J., Cui, X., Wang, Y., and Che, R.: Soil diazotrophic abundance, diversity, and community assembly mechanisms significantly differ between glacier riparian wetlands and their adjacent alpine meadows, Front. Microbiol., 13, https://doi.org/10.3389/fmicb.2022.1063027, 2022.

Gat, D., Mazar, Y., Cytryn, E., and Rudich, Y.: Origin-dependent variations in the atmospheric microbiome community in Eastern Mediterranean dust storms, Environ. Sci. Technol., 51, 6709-6718, https://doi.org/10.1021/acs.est.7b00362, 2017.

Jiang, X. Q., Wang, C. H., Guo, J. Y., Hou, J. H., Guo, X., Zhang, H. Y., Tan, J., Li, M., Li, X., and Zhu, H. Q.: Global meta-analysis of airborne bacterial communities and associations with anthropogenic activities, Environ. Sci. Technol., 56, 9891-9902, https://doi.org/10.1021/acs.est.1c07923, 2022.

Mazar, Y., Cytryn, E., Erel, Y., and Rudich, Y.: Effect of dust storms on the atmospheric microbiome in the Eastern Mediterranean, Environ. Sci. Technol., 50, 4194-4202, https://doi.org/10.1021/acs.est.5b06348, 2016.

*Ning, D., Yuan, M., Wu, L., Zhang, Y., Guo, X., Zhou, X., Yang, Y., Arkin, A. P., Firestone, M. K., and Zhou, J.: A quantitative framework reveals ecological drivers of grassland microbial community assembly in response to warming, Nature Communications, 11, 4717, https://doi.org/10.1038/s41467-020-18560-z, 2020.*

*Zhao, J., Jin, L., Wu, D., Xie, J., Li, J., Fu, X., Cong, Z., Fu, P., Zhang, Y., Luo, X., Feng, X., Zhang, G., Tiedje, J. M., and Li, X.: Global airborne bacterial community—Interactions with Earth's microbiomes and anthropogenic activities, Proc. Natl. Acad. Sci. U. S. A., 119, e2204465119, https://doi.org/10.1073/pnas.2204465119, 2022.*

**Comment 4**

Section 3.2 can be shortened. Some discussions/claims are not necessarily true due to the small dataset presented here. In another word, it is not your job to investigate the influence other possible factors. I would suggest putting more efforts on the diurnal difference of fungal aerosols which has been clearly demonstrated by the community structure and sugar tracers. It is very interesting and needs more data analysis to generate in-depth generalizations.

**Response:**

Thank you for the suggestions on the content of the manuscript. The corresponding information was added as follows:

Page 7, line 199-203: *"Fungal community structure exhibited distinct diurnal differences based on the results of NMDS analysis (Fig. S3). The Bray-Curtis dissimilarity index of fungal communities was slightly higher at night (0.46) than in the daytime (0.41), suggesting the high β diversity of fungal community at night. Co-occurrence network analysis also revealed that the average degree of fungal communities at night (10.0) was greater than in the daytime (8.9), suggesting that fungal taxa cooperated more closely with each other at night."*

Page 7, line 209-211: *"LEfSe analysis likewise shows clear diurnal differences in fungal taxa. The percentage of taxa belonging to Ascomycota increased significantly in the daytime, while taxa belonging to Basidiomycota dominate at night (Fig. S4)."*

Page 7-8, line 219-222: *"More fungal genera showed higher abundances at night, influenced by temperature and humidity (Table S3) (Reyes et al., 2016). Among the*

*dominant genera, the abundances of Coprinopsis, Coprinellus and Trametes were significantly higher at night than in the daytime (Coprinopsis: 29.9% > 6.9%, Coprinellus: 11.2% > 8.9% and Trametes: 7.7% > 5.8%).''*

The related analysis was added in Figure S4 and Table S3 as follows:

[Figure]

**Figure S4.** The cladogram of fungal taxa based on LEfSe analysis of fungal abundance between daytime and nighttime samples.

**Table S3.** The taxa at the phylum and genus levels with LDA scores > 2 and their relative abundances in the daytime and nighttime.

| Taxonomic level | Name | Day | Night | LDA (log$_{10}$) | Dominant group |
|---|---|---|---|---|---|
| phylum | Ascomycota | 65.3 | 28.6 | 5.3 | Day |
| genus | *Curvularia* | 0.64 | 0.11 | 3.4 | Day |
| genus | *Plenodomus* | 0.01 | 0.00 | 3.7 | Day |
| genus | *Periconia* | 1.86 | 0.26 | 3.9 | Day |
| genus | *Cladosporium* | 1.39 | 0.35 | 3.7 | Day |
| genus | *Sphaerulina* | 0.53 | 0.11 | 3.3 | Day |
| genus | *Aspergillus* | 2.88 | 0.26 | 4.1 | Day |
| genus | *Gelasinospora* | 0.19 | 0.02 | 3.0 | Day |
| genus | *Holtermanniella* | 0.46 | 0.16 | 3.2 | Day |
| phylum | Basidiomycota | 13.6 | 34.0 | 5.0 | Night |
| genus | *Phaeosphaeria* | 0.07 | 0.16 | 2.8 | Night |
| genus | *Dissoconium* | 0.01 | 0.08 | 2.8 | Night |
| genus | *Cyphellophora* | 0.00 | 0.04 | 2.9 | Night |
| genus | *Phyllopsora* | 0.00 | 0.02 | 3.3 | Night |
| genus | *Spathularia* | 0.00 | 0.02 | 3.1 | Night |
| genus | *Colletotrichum* | 0.02 | 0.12 | 2.9 | Night |
| genus | *Gibellulopsis* | 0.00 | 0.01 | 3.2 | Night |

| genus | Plectosphaerella | 0.08 | 1.49 | 3.8 | Night |
|-------|------------------|------|------|-----|-------|
| genus | Clonostachys | 0.02 | 0.20 | 3.1 | Night |
| genus | Lecanicillium | 0.02 | 0.04 | 2.9 | Night |
| genus | Cosmospora | 0.03 | 0.16 | 2.9 | Night |
| genus | Pseudocosmospora | 0.00 | 0.01 | 3.2 | Night |
| genus | Alfaria | 0.01 | 0.07 | 2.7 | Night |
| genus | Phomatospora | 0.01 | 0.07 | 2.7 | Night |
| genus | Phyllachora | 0.00 | 0.01 | 3.0 | Night |
| genus | Agaricus | 0.14 | 1.46 | 3.9 | Night |
| genus | Clitopilus | 0.01 | 0.11 | 2.8 | Night |
| genus | Crinipellis | 0.01 | 0.03 | 2.8 | Night |
| genus | Marasmius | 0.01 | 0.07 | 3.2 | Night |
| genus | Cryptomarasmius | 0.00 | 0.02 | 2.9 | Night |
| genus | Volvariella | 0.00 | 0.01 | 3.0 | Night |
| genus | Hohenbuehelia | 0.00 | 0.02 | 2.9 | Night |
| genus | Pterula | 0.00 | 0.00 | 3.2 | Night |
| genus | Aphanobasidium | 0.00 | 0.02 | 3.0 | Night |
| genus | Coprinopsis | 1.67 | 11.4 | 4.7 | Night |
| genus | Cristinia | 0.00 | 0.01 | 2.8 | Night |
| genus | Schizophyllum | 0.01 | 0.03 | 2.9 | Night |
| genus | Clitocybe | 0.02 | 0.16 | 2.9 | Night |
| genus | Melanoleuca | 0.00 | 0.02 | 3.0 | Night |
| genus | Coniophora | 0.00 | 0.01 | 3.5 | Night |
| genus | Ceratobasidium | 0.00 | 0.02 | 3.3 | Night |
| genus | Limonomyces | 0.07 | 1.35 | 3.8 | Night |
| genus | Hyphodontia | 0.00 | 0.03 | 2.7 | Night |
| genus | Hydnophlebia | 0.04 | 0.21 | 3.0 | Night |
| genus | Hyphoderma | 0.04 | 0.13 | 2.8 | Night |
| genus | Brunneoporus | 0.01 | 0.02 | 3.0 | Night |
| genus | Podoscypha | 0.00 | 0.02 | 3.0 | Night |
| genus | Peniophora | 0.00 | 0.25 | 3.1 | Night |
| genus | Trechispora | 0.07 | 0.08 | 2.6 | Night |
| genus | Kondoa | 0.00 | 0.06 | 2.8 | Night |
| genus | Golubevia | 0.00 | 0.01 | 2.9 | Night |
| genus | Meira | 0.00 | 0.02 | 2.9 | Night |
| genus | Rhodosporidiobolus | 0.02 | 0.05 | 3.1 | Night |
| genus | Sporobolomyces | 0.02 | 0.15 | 3.0 | Night |
| genus | Colacogloea | 0.00 | 0.02 | 3.3 | Night |
| genus | Hannaella | 0.01 | 0.09 | 2.8 | Night |
| genus | Bulleromyces | 0.01 | 0.03 | 3.0 | Night |

**Comment 5**

In general, I would suggest shortening the part of results which only presents data and describes the variation. More importantly, Section 3.4.2 should be polished with the

emphasis on the used method and details of interaction. I do not fully understand how the interaction happened. How about the sensitivity of the interaction? For example, did the nighttime processes give more weight on the interaction?

**Response:**

Thank you for the suggestions on the content of the manuscript. The results were shortened as suggested. To better understand how the interaction happened, the content about network construction was added as follows:

[revised manuscript text omitted]

Thank you very much for your comments and suggestions. Your any further comments and suggestions are appreciated.